# Unusual Histology in Mesothelioma: A Report of Two Cases with a Brief Review

**DOI:** 10.3390/diagnostics12020371

**Published:** 2022-02-01

**Authors:** Francesca Bono, Stefano Ceola, Carlo Beretta, Marta Jaconi

**Affiliations:** 1Pathology Unit, San Gerardo Hospital, 20900 Monza, Italy; m.jaconi@asst-monza.it; 2Department of Medicine and Surgery, Pathology, San Gerardo Hospital, University of Milano-Bicocca, 20900 Monza, Italy; s.ceola1@campus.unimib.it (S.C.); c.beretta36@campus.unimib.it (C.B.)

**Keywords:** mesothelioma, adenomatoid mesothelioma, signet ring cell mesothelioma, unusual variant epithelioid mesothelioma

## Abstract

Mesothelioma is often difficult to diagnose due to its rarity and its unusual histopathological features that could lend to diagnostic pitfalls and misdiagnosis. The WHO histological classification of pleural tumors in 2021 recommended a pathologic grading system for malignant pleural mesothelioma. Architectural aspects and cytological features, with nuclear grading, bent on a neoplastic score with fundamental prognostic and diagnostic value. Unusual features must be correctly assigned in the grading system to avoid misdiagnosis, especially toward metastatic lesions or reactive pleural processes. In this paper, we present two cases as examples of unusual morphological and architectural features with a brief literature review.

## 1. Introduction

In 2021, WHO published its latest volume about thoracic tumors. Pleural diffuse mesothelioma in this issue retains the three major histological subtypes as previously described (epithelioid, sarcomatoid and biphasic) and more focus is placed on architectural patterns and cytological and stromal features [1].

In the case of epithelioid mesothelioma, architectural patterns are underlined, as in previous editions, as well as cytological features and stromal appearance, due to their prognostic value and to avoid possible diagnostic pitfalls. Epithelioid mesothelioma represented approximately 80% of all pleural mesotheliomas. It is defined as being composed of deceptively bland, uniform cuboidal (epithelioid) cells that usually infiltrate the pleura in a tubulo-papillary growth pattern, consisting of round-to-oval structures admixed with tumor cells covering a fibrovascular core. However, a wide range of architectural growth patterns and cytologic features may be encountered. Therefore, in the histological report of epithelioid mesothelioma, it is suggested the indication of grade (if high or low), all architectural patterns present with their percentages, and the surgical type of resection, if it is a definitive resection (extrapleural pneumonectomy or extended pleurectomy/decortication).

Regarding architectural patterns, tubulopapillary, trabecular and adenomatoid are considered prognostically favorable; solid growth in more than 50% of the tumor and micropapillary pattern are instead unfavorable.

The cytological features to report are also divided prognostically as more favorable (lymphohistiocytoid cells and low nuclear grade) and unfavorable (rhabdoid cells, pleomorphic cells, and high nuclear grade). The nuclear grading of pleural diffuse epithelioid mesothelioma is newly introduced in this latest edition and stratifies epithelioid mesothelioma in “overall tumor grade high” and “low” using nuclear atypia (with 1 point for mild, 2 for moderate, and 3 for severe) and mitotic count (with 1 point for low, ≤1 mitosis/2 mm^2^; 2 points for intermediate, 2–4 mitoses/2 mm^2^; and 3 points for high, ≥5 mitoses/2 mm^2^). This change is due to the fact that grading, despite not being previously recommended for mesotheliomas, has demonstrated prognostic significance [2,3,4].

Nuclear grading, combined with the presence or absence of necrosis, results in an overall tumor score that is defined as low for nuclear grades I and II without necrosis, and high in tumors that show nuclear grade II with the presence of necrosis or nuclear grade III with or without necrosis [5].

Lastly, stromal features were suggested to be included in the report because, if predominantly myxoid, predict a better prognosis [6,7]. In epithelioid mesotheliomas, the most difficult challenges in differential diagnoses are posed by reactive processes [8].

In epithelioid mesotheliomas, the most difficult challenges in differential diagnoses are posed by reactive processes [3].

Such differential diagnosis is in many cases particularly hard, but some architectural features, such as the demonstration of fat infiltrations or molecular alterations (e.g., BAP1 inactivating mutations), help to achieve this goal [4,5].

Furthermore, some morphological features, due to their rarity, make differential diagnosis more challenging even with possible metastatic lesions [9,10].

In this paper, we present two cases as examples of unusual morphological and architectural features, and a brief review of analogue cases reported, with some possible solutions.

## 2. Case N°1

A 68-year-old male patient with possible exposure to asbestos came to our attention for chest pain. He underwent an X-ray scan and abdominal ultrasound, which revealed a right pleural effusion. A TC scan was performed, and pleural effusion was confirmed, with the additional reporting of enlargement of the right pleural membranes. 

The patient then underwent a right thoracoscopy, pleural biopsies and DRAINAPORT placement.

The histological examination showed pleural infiltration by epithelioid cells arranged in tubular and cystic spaces with adenomatoid features. The pleura was infiltrated in its entire thickness, and a focal invasion of fat was identified. Some mitoses were observed; no necrosis was observed. Immunohistochemical reactions demonstrated the mesothelial origin of the neoplastic cells, with positivity for WT1, calretinin and pancytokeratin; TTF1, CDX2 and CK20 were negative.

Immunostaining for BAP1 was performed, with retained nuclear expression (Figure 1).

A diagnosis of epithelioid mesothelioma, of the adenomatoid type, was formulated and the patient underwent a first line of therapy with cisplatin/pemetrexed, followed by subtotal right pleurectomy 6 months later. Six months later, a radiological progression was observed, with contralateral involvement, accompanied by cough and worsening of the dyspnea. The patient was therefore hospitalized, and he was subjected to left pleurodesis biopsies and was diagnosed with epithelioid pleural mesothelioma. Two months later, the patient eventually died of his disease.

## 3. Case N°2

A 53-year-old male railway worker, former smoker with exposure to asbestos, came to clinical observation for dyspnea that had been worsening for more than one year. 

An X-ray scan of the thorax was performed, which revealed an extensive unilateral pleural effusion. The cytological examination showed mildly atypical mesothelial cells (WT1+/calretinin+) with loss of BAP1 nuclear expression. A CT scan was therefore performed, and pointed out the extensive enlargement of the diaphragmatic pleura, with involvement of the intercostal muscles.

Subsequently, a surgical diagnostic biopsy (VATS) was performed and the histological examination revealed infiltration of the fat and pleura by epithelioid cells growing in a diffuse pattern, with vacuolated cytoplasm and signet ring appearance (Figure 2). Immunohistochemical reactions demonstrated the mesothelial nature of the neoplastic proliferation, with positivity for mesothelial markers (WT1, calretinin, and D2-40) and loss of BAP1 nuclear expression.

The patient underwent a first line chemotherapy with cisplatin and pemetrexed and was then subjected to a right pleural decortication, with the partial removal of diaphragm and pericardium, followed by RT (45 Gy in 25 fractions).

At the time of writing of this article, 18 months later, the patient is alive and the disease is under control.

## 4. Discussion

Mesothelioma is often difficult to diagnose due to its rarity and its unusual histopathological features, which could lend to diagnostic pitfalls and misdiagnosis. 

Mesothelioma in its epithelioid appearance must be distinguished from carcinomas and other epithelioid neoplasms that can infiltrate the pleura in a diffuse way, with a pseudomesotheliomatous appearance. Most commonly, these neoplasms are carcinomas, and particularly those originating from the lung, breast, ovary and gastrointestinal tract, but also kidney, prostate and pancreas malignancies can potentially give metastases to the pleura [1]. Tumors with epithelioid morphology should also be considered, particularly epithelioid vascular malignancies, epithelioid sarcoma, and melanoma [11]. Clinical and radiological features may be of some help because mesothelioma characteristically presents as multiple pleural nodules or diffuse pleural thickening; other tumors involving the pleura may mimic this growth pattern, but the finding of large or multiple lung parenchymal masses should raise concerns against a mesothelioma diagnosis [12]. Sometimes, it could be even more difficult to discriminate between epithelioid mesothelioma and some benign processes, such as reactive mesothelial hyperplasia, particularly when it is marked in some infection or collagen vascular disease or when an invasive component in the adipose tissue or chest wall cannot be demonstrated.

Since the morphological features can be misleading, ancillary studies are needed. 

A useful help can come from molecular findings; in particular, the most frequent altered genes in mesothelioma are BRCA1-associated protein 1 gene (BAP1), cyclin-dependent kinase inhibitor 2A (CDKN2A,) and neurofibromatosis type 2 gene (NF2), coding for merlin, an oncosuppressor protein involved in mTOR signaling [1,2,3,4]. Combined mutations of BAP1, NF2, and CDKN2A are observed in about 34% of malignant mesothelioma, indicating the importance of these tumor suppressors in the disease pathogenesis [13]. Indeed, it has been observed that the combined deletion of BAP1, NF2, and CDKN2A genes causes rapid disease onset in mice [14]. Recently, Zhang et al. has described a detailed picture of the mesothelioma genomic landscape with an exon sequencing approach. Their work has shown how mesotheliomas mostly follow a linear evolution with BAP1 being the most frequent ancestral mutation and NF2 arising mainly as a late event. Moreover, a minority of patients presented a branched evolution that was associated with a higher tumor lymphocyte infiltration and mutational burden, suggesting a possible therapy approach with immunomodulators [15]. An aberrant copy number of alterations in CDKN2A identified with sequencing approaches were confirmed in other studies through fluorescent in situ hybridization (FISH) and IHC, and they were associated with higher asbestos fiber exposure [16]. 

As reported above, the BAP1 gene mutation is the most frequent genomic alteration in malignant pleural mesothelioma; BAP1 orchestrates chromatin-associated processes, including gene expression, DNA replication and DNA repair. Somatic bi-allele mutation resulting in BAP1 loss has been associated with improving prognosis and it has been observed in nearly 60% of malignant pleural mesothelioma [17] in uveal melanoma and other tumors [18]. BAP1 was also recently found to be frequently mutated in 25% of malignant peritoneal mesothelioma, a rare cancer usually not associated with asbestos exposure [19], but more often associated with the mutation of other chromatin modifiers, indicating the importance of epigenetic regulation in mesothelioma pathogenesis. It has also to be mentioned how the loss of BAP1 protein expression in several cancers can be observed in wild-type genes [20]. There have been reports of several cases of malignant pleural mesothelioma with no BAP1 gene mutation and with normal mRNA expression, which are negative for BAP1 protein staining, so post-translational events regulating instability might be responsible for BAP1 loss of function in such cases [21].

The use of a specific immunohistochemistry panel for detecting these alterations should be performed depending on, and in combination with, cytoarchitectural characteristics, clinical and radiological findings, to address the possible differential diagnosis (i.e., to distinguish an epithelioid mesothelioma with clear cell features from a clear cell renal carcinoma) [22,23].

BAP1 defective expression is routinely used to aid in distinguishing mesothelioma from reactive mesothelial proliferations, with a relatively low sensitivity (42–65%), but a specificity of 100% [24].

Analogue use is made of MTAP that can be detected by immunohistochemistry; its nuclear expression is lost in association to the loss of CDKNA 9p21 encoding p16, with a stackable sensitivity (42–48%) and absolute specificity (100%) [9,25,26,27,28].

The loss of immunohistochemical expression of BAP1 helps to make the correct diagnosis in many cases in which the differential diagnosis is between epithelioid mesothelioma and reactive hyperplasia. Meanwhile, the loss of BAP1 is not useful in cases in which the alternative diagnosis is another epithelial neoplasm [29].

In the same way, immunohistochemistry for MTAP or homozygous deletion of 9p21 (CDKN2A) in FISH is useful mainly in sarcomatoid mesothelioma that exhibit these losses in >80% of cases [30].

Furthermore, the combined use of both staining methods contributes to increase the reliability of the diagnosis.

These considerations and molecular alterations, demonstrated by BAP1 loss in our case N°2 (signet ring cell mesothelioma), were not applicable to case N°1, in which nuclear immunostaining for BAP1 was retained. In this particular case, however, the peculiar architectural way of growth and infiltration along the entire thickness of the pleura supports the diagnosis of malignant mesothelioma, of the adenomatoid variant. This peculiar variant, so called for its morphological similarity with adenomatoid tumors of the genital tracts as reported by Weissferdt, Khalor and Suster [31], must be differentiated not only from benign pleural adenomatoid tumor (an extremely rare lesion, with <20 cases reported in the literature), but also from metastatic adenocarcinoma and epithelioid hemangioendothelioma. The positivity of mesothelial markers, such as WT1, calretinin, podoplanin and HBME1, along with clinical–radiological findings, helps to distinguish this peculiar variant of malignant mesothelioma from a metastatic adenocarcinoma from another site. Hemangioendothelioma could instead give more trouble, but clinical data, as younger age at presentation, female predilection, and multivisceral involvement, could avoid misdiagnosis, associated with the expression of vascular markers, such CD31, CD34 and Factor VIII positivity, accompanied by negativity for mesothelial markers, such as WT1 and calretinin [31].

In case N°2, we report a signet ring cell mesothelioma, described for the first time in 2003 by Cook et al. [32], and with just over 20 cases reported in the literature in English [33]. Clinically, these peculiar cytological features do not lead a different clinical course; additionally, signet ring cell mesothelioma maintains a predominance for pleural site versus peritoneal location, a predominance of male patients and is associated with exposure to asbestos. The prognosis of signet ring cell mesothelioma is equally poor, with 15 months of median survival starting from diagnosis. This peculiar cytological feature makes the differential diagnosis challenging, especially towards metastatic adenocarcinoma, signet ring type, and more so in peritoneal sites or in the case of aberrant expression of CDX2 or cytokeratin 20, as reported by Rossi et al. [23]. Furthermore, rare cases of ALK rearranged mesotheliomas and signet ring cell adenocarcinoma from the lung ALK-rearranged are also included in differential diagnoses [34] 

Moreover, unusual clinical–radiological findings and immunostaining patterns could be a diagnostic pitfall, even with the usual morphological features of malignant mesothelioma [23].

We can conclude that malignant mesothelioma is an uncommon lesion; its detection could be frequently troublesome and may be hard to distinguish from hyperplastic reactive processes and metastatic lesions. 

Ultimately, the two cases presented in this paper are merely examples of some diagnostic difficulties encountered in the management of the pathology of malignant epithelioid mesotheliomas, but with some ancillary methods, as discussed before, a correct diagnosis is possible, bearing in mind also the rare histological variations. Furthermore, a diagnostic pitfall is not only related to morpho-architectural features, but we should also keep in mind the clinical, radiological and unusual patterns of immunostaining that sometimes are misleading even if they present with the usual morphological features [23].

## Figures and Tables

**Figure 1 diagnostics-12-00371-f001:**
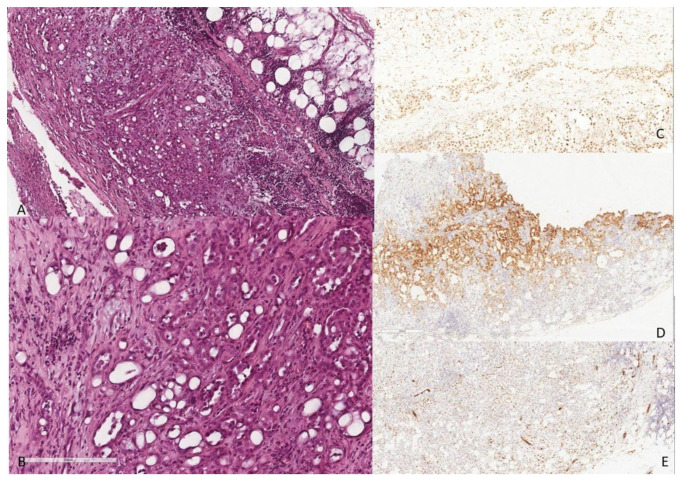
Infiltration of the entire thickness of the pleura with focal fat invasion by epithelioid cells arranged in tubular and cystic spaces with adenomatoid features ((**A**) H&E, ×100). At higher magnification, the cytological features of tumor cells can be appreciated: round nuclei, moderate amounts of eosinophilic cytoplasm and conspicuous nucleoli; some mitoses are present ((**B**) H&E, ×400). Immunohistochemical reactions demonstrated the mesothelial origin of the neoplastic cells ((**C**) PAB1 retained nuclear expression, ×400; (**D**) calretinin cytoplasmic stain, ×400; (**E**) WT1 nuclear stain, ×400).

**Figure 2 diagnostics-12-00371-f002:**
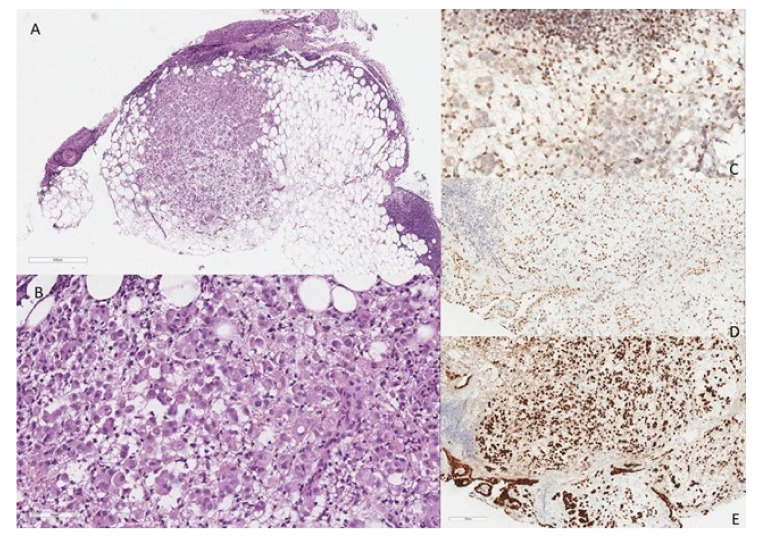
Epithelioid cells growing in a diffuse pattern, with vacuolated cytoplasm and signet ring appearance ((**A**) H&E ×40). The higher magnification better demonstrates the morphologic features of the signet ring cells with pycnotic crescent shaped nuclei located at cell periphery ((**B**) H&E, ×400). Immunohistochemical reactions demonstrated the mesothelial nature of the neoplastic proliferation ((**C**) PAB1 loss of nuclear expression, ×400; (**D**) WT1 nuclear stain, ×400; (**E**) calretinin cytoplasmic stain, ×400).

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
