# Peer review of "Unusual Histology in Mesothelioma: A Report of Two Cases with a Brief Review"

_diagnostics, 2022, doi:10.3390/diagnostics12020371_

Round 1

Reviewer 1 Report

In this manuscript the authors present a report of two cases of mesothelioma with uncommon histology features. Mesothelioma is a rare malignancy. Unfortunately, current diagnostic approaches might result in misdiagnosis. Although, this is only a report of two cases, the authors discuss well their observations according to the current literature on the same field.

Author Response

Thank you very much for your review.

Please see attachment for the revised version with corrections.

kind regards

Francesca Bono

Reviewer 2 Report

The ms proposes the discussion of two italian cases of malignant mesothelioma. The ms suffers of some english errors and other mistakes (capitalized words, bold, different fonts, etc). There is no data about possible asbestos exposure for cases. Discussion is very confusing not focused on the two cases.

Author Response

Dear reviser,

please see attachment for corrected version. As mentioned in the text, asbesto's exposure was confirmed in the case 2, suspected in the case 1.

Kind regards

Francesca Bono

Reviewer 3 Report

As the authors pointed out, mesothelioma is difficult to make diagnosis sometimes due to its unique histological features. It would be helpful to evaluate cytology including nuclear grading and tumor specific markers.

This manuscript presented two mesothelioma cases with literature review. Overall, this work is well-done and could be publishable if a minor revision could be made.

Minor comments:

Line 11-12, Architectural aspects and cytological features, with nuclear grading, bent on a neoplastic score with fondamental prognostic and diagnostic value. Please check the spelling.

Line 65-66, A TC scan was performed and pleural effusion was confirmed, with the additional reporting of enlargement of right pleural membranes. “TC” must be wrong, please correct.

Line 81-82 and line 114-115, Figure 1&2 legends: Immunohistochemical reactions demonstrated mesothelial origin of the neoplastic cells (C, PAB1 retained nuclear expression x400;  “PAB1” should be BAP1.

Fig 1C-E, nuclear staining for WT1, it would be better to replace the image with higher resolution. Pease also to confirm the images are high magnification 400x, it looks quite small.

Mesothelin is also considered as an important marker while making diagnosis.  By the way, did you look at the expression of mesothelin in the cases?

CT scan was performed in both cases, thus it would be interesting to include the CT scan images if possible.

Author Response

As the authors pointed out, mesothelioma is difficult to make diagnosis sometimes due to its unique histological features. It would be helpful to evaluate cytology including nuclear grading and tumor specific markers.

This manuscript presented two mesothelioma cases with literature review. Overall, this work is well-done and could be publishable if a minor revision could be made.

 Minor comments:

Line 11-12, Architectural aspects and cytological features, with nuclear grading, bent on a neoplastic score with fondamental prognostic and diagnostic value. Please check the spelling.

Response:  To be replaced with: as in previous editions is underlined the importance of architectural patterns, cytological features and stromal appearance due to their prognostic value and to avoid possible diagnostic pitfalls

Line 65-66, A TC scan was performed and pleural effusion was confirmed, with the additional reporting of enlargement of right pleural membranes. “TC” must be wrong, please correct.

Response:  to be corrected to CT scan

Line 81-82 and line 114-115, Figure 1&2 legends: Immunohistochemical reactions demonstrated mesothelial origin of the neoplastic cells (C, PAB1 retained nuclear expression x400;  “PAB1” should be BAP1.

Response:  to be corrected to BAP1

Fig 1C-E, nuclear staining for WT1, it would be better to replace the image with higher resolution. Pease also to confirm the images are high magnification 400x, it looks quite small.

Response: please see new corrected  version of IMAGE 1 in attached file

Mesothelin is also considered as an important marker while making diagnosis.  By the way, did you look at the expression of mesothelin in the cases?

Response: Unfortunately Mesothelin is not performed by our laboratory.

CT scan was performed in both cases, thus it would be interesting to include the CT scan images if possible.

Response: please see attached as file jpg images of CT scan from both cases.

Round 2

Reviewer 2 Report

The authors did not modify the discussion section, as suggested.

I cannot propose the acceptance of the ms.

Response: 

I propose to insert at line nr 248 this section to clarify the discussion: “Ultimately, the two cases here presented are merely examples of some diagnostic difficulties encountered into pathologic managing malignant epithelioid mesothelioma, but with some ancillary aims, as discussed before, the correct diagnosis is possible, bearing in mind also these rare histological variations.”

Kind regards

Francesca Bono

Round 3

Reviewer 2 Report

Dear Ms. Lareina Guo, Section Managing Editor, M.Sc. I confirm the acceptance of the manuscript. Sincerely yours   Simona

This manuscript is a resubmission of an earlier submission. The following is a list of the peer review reports and author responses from that submission.